# Neuropsychological Alterations of Prolactinomas’ Cognitive Flexibility in Task Switching

**DOI:** 10.3390/brainsci13010082

**Published:** 2023-01-01

**Authors:** Chenglong Cao, Wen Wen, Aobo Chen, Shuochen Wang, Guozheng Xu, Chaoshi Niu, Jian Song

**Affiliations:** 1Department of Neurosurgery, The First Affiliated Hospital of USTC, Division of Life Sciences and Medicine, University of Science and Technology of China, Hefei 230001, China; 2Department of Cognitive Neuroscience, Faculty of Psychology & Neuroscience, Maastricht University, P.O. Box 616, 6200 MD Maastricht, The Netherlands; 3Department of Neurosurgery, The General Hospital of Chinese PLA Central Theater Command, Wuhan 430074, China; 4Department of Psychological & Brain Sciences, Boston University, Boston, MA 02215, USA

**Keywords:** prolactinomas, cognitive flexibility, theta oscillation, machine-learning, connectivity

## Abstract

Prolactinomas have been reported to impair cognition in broad aspects. However, few studies investigated the influence of prolactinomas on cognitive flexibility never mentioning the underlying neural and electrophysiological mechanism. We recorded scalp electroencephalography (EEG) in a colour-shape switching task. Patients with prolactinomas showed longer reaction time in switch trials and larger switch costs relative to healthy controls (HCs). Compared to HCs who showed stronger frontal theta activity in switch trials, the generally weak frontal theta activity in patients implied that they could not afford the executive control to configure task sets. Meanwhile, machine-learning based classification revealed that patients manifested non-selective brain patterns in response to different task types (colour vs. shape task) and different task states (switch vs. repeat state), which collectively suggested the cognitive dysfunction in preparation for a changing environment. Compared to HCs who showed stronger frontoparietal synchronization in switch trials, this enhanced frontoparietal connectivity was disrupted among patients with severe prolactinomas. This finding implicated greater hyperprolactinemia was linked to a larger decrease in cognitive performance. Taken together, the present study highlighted frontal theta power, and frontoparietal connectivity at theta band as the electrophysiological markers of the impaired cognitive flexibility and task control in patients with prolactinomas.

## 1. Introduction

Pituitary adenomas (PAs) are the second most common intracranial tumours, accounting for around 16.5 percent of central nervous system tumours [1]. Prolactinomas are the most prevalent subtype of pituitary tumours [2] and are typically characterised by hypersecretion of prolactin (PRL) in the circulating blood suppressing the release of sex steroid hormones [3]. Despite the significant decline of PAs’ mortality through endocrinological and surgical treatments, emerging studies raised the concern to examine cognitive dysfunctions in patients with Pas [4] including decreased attention [5,6], impaired working memory [7], emotion processing [8], and inhibitory controls [9,10,11,12]. In particular, present understanding of pituitary tumours’ influence on executive functions and the underlying neural mechanism is still limited.

Executive functions refer to a collection of cognitive operations such as response inhibition and interference control in selective attention, working memory and decision making [13], which enable humans to rapidly adapt to the ever-changing environment. Previous studies investigated executive control ability of patients with prolactinomas using the Go/Nogo paradigm and demonstrated that patients showed impaired response inhibition [9,10,11,12]. Besides response inhibition, a full expression of executive functions should also involve accomplishing goal-orientated missions according to task rules and switching between tasks. As a crucial component of executive functions, the capacity to switch between activities is particularly associated with the cognitive flexibility to deactivate the task set that was relevant on the previous trial and activate the task set that is currently relevant [14]. Switch costs are observed as the discrepancies in speed and/or accuracy as a result of the varying demands imposed on executive control during task switching [15]. Hence, switch costs represent the increased demand for executive control caused by the necessity to set up the new configuration while dealing with enormous interference produced by the old task [16]. 

In a conventional task switching paradigm, event-related neural activities in the cue interval allow the investigation of proactive control in preparation for forthcoming tasks. Theta oscillations, in particular, reflect the neural process in enrolling cognitive control associated with executive functions [17]. Stronger theta activities at the prefrontal and posterior areas were observed in switch trials compared to repeat trials [18]. Additionally, long-range communications mediated by theta-band activity serve as a messenger in general. Previous studies demonstrate that task switching engages frontoparietal networks as evidenced by enhanced inter-site phase synchronization [19,20,21]. 

Broad regions are activated in task switching, including the dorsolateral and ventrolateral prefrontal cortex, the supplementary and pre-supplementary motor regions, and the superior and inferior parietal areas [22,23]. Frontal and parietal brain regions in particular support flexible task representations and rule discrimination under varying external demands [24]. A recent structural MRI study demonstrates decreased grey matter volume of frontal cortex in prolactinomas [25] but has not been able to link the functional relevance of frontal cortex impairment with cognitive deficits. Given the alteration of anatomical structures in prolactinomas, we hypothesize larger switch costs due to their deteriorated cognitive flexibility. With these in mind, we aimed to examine the cognitive as well as neural oscillatory changes of patients with prolactinomas using the task switching paradigm.

## 2. Method

### 2.1. Participants

Prolactinomas patients were recruited in the Department of Neurosurgery, Wuhan School of Clinical Medicine and General Hospital of Northern Theatre Command. Inclusion criteria were the same as previous research [11]: (1) patients had a prolactin-secreting pituitary tumour [26] and were resistant to long-term medication therapy, (2) they had no history of radiation therapy or craniotomy, (3) they had a normal or corrected-to-normal vision and hearing, and (4) they were able to complete EEG tests. Patients were excluded if they met any of the following criteria: (1) had a history of neurologic or psychiatric disorders; (2) had comorbidities that could impair cognitive function, such as severe liver, heart, or kidney dysfunction; (3) had severe complications, such as comas, infections, epilepsy, hydrocephalus, and leaking cerebrospinal fluid; and (4) had drug or alcohol abuse [subjects who drink alcohol over 2.0 standard drinks (10 g of pure alcohol) during the day and meet any 2 of the 11 criteria under the DSM-V in the past year] [27], or were on any medications (including oral contraceptives and dopamine agonists). To account for diurnal fluctuations in hormone levels, venous blood samples were taken between 8:00 and 9:30 a.m. Since this study targeted at prolactinomas, we measured serum PRL(ng/ml), which was determined by chemiluminescent immunoassays (Roche, cobas^®^ 8000, Switzerland). The maximum PRL concentration is 208 ng/ml because PRL concentration exceeding 208ng/ml is not of clinical diagnostic significance. Studies have shown abnormalities in the brain structure of pituitary patients induced by macroadenomas [28]. The patient group in our study was carefully chosen to exclude large tumours compressing optic nerves or adjacent brain regions and to avoid potential bias of our findings. 

Twenty-eight patients and thirty-two healthy control adults participated in the study. The healthy controls (HCs) were recruited from healthy volunteers with matched age and education levels. Participants who had chance-level behavioural performance (one patient and three HCs) or noisy EEG data (missing trials or extensive noise lead to <70% clean trials remained, one patient and three HCs) were excluded from further analysis. The final sample consisted of 26 patients and 26 healthy controls (see Table 1). The study was approved by the ethics committee of Wuhan School of Clinical Medicine, Southern Medical University ([2018]003-1). The written informed consent was explained carefully and obtained from all participants.

### 2.2. Stimuli and Procedure

The experiment was carried out in a sound-attenuated and dimly lit room. Participants were seated in a comfortable chair while performing a colour-shape task. The experiment was built on E-prime 3.0 software (Psychology Software Tools, Pittsburgh, PA). Each trial starts with an instructional cue (Chinese character of ‘colour’ or ‘shape’) indicating whether the shape task or the colour task is relevant for the current trial. The task cue was presented for 1500 ms. In the colour task, participants were instructed to judge whether the target stimuli were blue (right-click of a computer mouse) or red (left-click). In the shape task, participants were instructed to judge whether the task is a "circle" ("right" click) or a "square" ("left" click). The target stimulus was shown for 1000 ms (see Figure 1). Trials were defined as a repeat/switch trial if the current task is the same as/different from the previous trial. There were 201 trials in total (the first initiation trial was removed from analysis) and half of them were switch trials. Participants had a break after completing 67 trials. Participants were fully practised prior to the formal experiment. 

### 2.3. EEG Recording

The EEG was recorded using a 32-channel cap based on the international 10–20 system (eegoTM amplifier, Germany). CPz was the online reference electrode. The impedance of the EEG recording was kept below 5 KΩ. Continuous EEG signals were obtained with a bandpass of 0.05–200 Hz at a sample rate of 1000 Hz.

### 2.4. Data Analysis

Behaviour. The main behavioural dependent variable of interest was reaction time (RT) calculated based on correct trials. Outlies beyond 2.5 standard deviations of mean were removed. Mixed two-way ANOVA with group (Patients vs. HCs) as the between-subject factor and condition (repeat vs. switch) as the within-subject factor was performed on mean RT and accuracy.

Preprocessing. Offline preprocessing was performed using EEGLAB [29]. We bandpass-filtered the data to 1–40 Hz and re-referenced the data to the averaged mastoids. Bad channels were spherically interpolated. Independent Component Analysis (ICA) was performed to correct blinks, ocular movements, and other artefacts. There were 5.3 (SD = 1.8) and 5.3 (SD = 2.1) components being rejected for HCs and patients, respectively. The continuous EEG signal was then segmented into different epochs depending on the specific analysis. Epochs containing EEG signals beyong the threshold ±60 μV were excluded. Improbable and abnormally distributed data beyond 8 standard deviations of the mean probability distribution and kurtosis distribution were also removed. These resulted in 4% and 6% trials removed among HCs and patients.

Multivariate decoding. To examine the brain activity under different task types (shape vs. colour) and task states (switch vs. repeat) in preparation of upcoming targets, machine-learning-based classification was performed on the broad-band EEG data during the cue interval. The continuous EEG was segmented into −0.2-1.5 s timelocked to the cue onset and baseline-corrected to the mean amplitude of the pre-cue interval. We Gaussian-smoothed (window size = 8 ms) the data along the time dimension to reduce temporal noises. Cross-validation was carried out in a leave-one-fold-out manner. At each time point, data was partitioned into nine training folds and one test fold. In the task decoding, trials were labelled based on the current task type (shape vs. colour). In the state decoding, trials were labelled based on task state (switch vs. repeat). Equal number of trials of each condition (e.g., switch and repeat trials) were used to train the classifier through subsampling. We first estimated the covariance matrix based on the trial-averaged training set of the same condition. Next, we computed the pairwise Mahalanobis distances between test trials and the trial-averaged training set using the covariance matrix. A smaller distance toward the trial-averaged activation indicated a larger pattern similarity. The decoding was marked as a success when the distance between this given trial and the activation profile of the same condition was smaller than the other activation profile in the training data. This routine was iterated 1000 times and trials were randomly assigned to the training and test set in each iteration. Decoding accuracy across iterations, trials, and time samples was averaged to represent the classification performance of each individual. Since the outcome of binary classification is either above or at chance level (50%), a one-sample *t*-test (upper-tail, alpha = 0.05) was performed for statistics at group-level.

Time-frequency analysis. Time-frequency decomposition was conducted using Fieldtrip [30] and customized Matlab scripts (MathWorks, Natick, MA, USA). Epochs were extracted from −2.5–3 s relative to cue onset. Extra time points were included to avoid the edge effect [31]. After the time-frequency decomposition, we selected data between −0.3 and 1.5 s for further statistical testing. The surface Laplacian filtering was performed to attenuate the volume conduction. The averaged ERP was subtracted from each trial to remove the phase-locked activity. We used the Morlet wavelet convolution with a kernel width linearly increase from 2 to 7 to extract the time-frequency activity with a 50 ms step. The power value of each trial was normalized to the pre-stimulus baseline (−0.3–−0.1 s) using decibel conversion. Task-related theta (2–7 Hz) activity at frontal electrode Fz was assessed by comparing switch and repeat trials in patients and HCs. Nonparametric cluster-based permutation was performed on the theta power time-series over participants (alpha = 0.05, cluster-based nonparametric alpha = 0.05, cluster statistic = sum, two-tail, permutation times = 10,000 [32]). Interareal connectivity was measured using phase-lock-value (PLV). Driven by the theoretical hypothesis as well as the topographical distribution of connectivity referenced to Pz, we select F3 and FC1 as the representative of frontal cluster.

## 3. Results

### 3.1. Behaviour

Mixed two-way ANOVA demonstrated that participants were generally slower in switch trials (F(1, 50) = 27.122, *p* < 0.001, η_p_^2^= 0.352) and patients were slower than HCs (F(1, 50) = 4.525, *p* = 0.038, η_p_^2^= 0.083). Although patients showed larger switch cost (31.7 ms) than HCs (15.5 ms) in reaction time, the interaction effect just reached marginal significance (F(1, 50) = 3.209, *p* = 0.079, η_p_^2^= 0.060). Analysis of accuracy revealed a main effect of trial conditions (F(1, 50) = 8.683, *p* = 0.005, η_p_^2^= 0.148), such that participants exhibited lower accuracy in switch trials (84.4%) than repeat trials (86.7%)(see Figure 2). There was no significant difference between groups (F(1, 50) = 1.918, *p* = 0.172, η_p_^2^= 0.037) nor significant interaction between group and trial condition (F(1, 50) = 0.202, *p* = 0.655, η_p_^2^= 0.004).

### 3.2. Non-Selective Preparatory Brain States in Patients with Prolactinomas 

Figure 3A illustrates the result of Mahalanobis-distance based task type decoding (colour vs. shape task) of two groups. Classification accuracy of HCs was significantly higher than chance level (*t*(25) = 1.718, *p* = 0.049, Cohen’s *d* = 0.337) while classification accuracies of patients were barely above chance (*t*(25) = 0.592, *p* = 0.720, Cohen’s *d* = 0.116). This suggests that HCs manifested distinctive brain activities in the colour task and the shape task. Analogously, we labelled trials based on whether task-state switching was needed. Classification of switch trials over repeat trials was successful in HCs (Figure 3B, *t*(25) = 2.396, *p* = 0.012, Cohen’s *d* = 0.470) but not in patients (*t*(25) = 0.506, *p* = 0.309, Cohen’s *d* = 0.099). These implied that patients’ deterioration in instantiating selective preparatory task states.

### 3.3. Patients Showed Decreased Frontal Theta Power

Next, we examined the oscillatory activity during cue interval at frontal electrode (Fz). There was a prominent non-phase-locked rhythmic activity at theta band (Figure 4A). To further compare the frontal theta activity in switch and repeat trials, we evaluated the time-series of theta power in each group (Figure 4B) and found that only HCs exhibited a significant theta power increase in switch trials (0.55–0.95 s, clustered *p* = 0.023). By contrast, theta activity was generally weak in patients. The functional relevance of theta activity was reflected by the negative correlation between theta activity and switch cost, such that participants with larger frontal theta power difference between switch and repeat trials would have smaller switch costs (Figure 4C).

### 3.4. Prolactinomas Impaired Frontoparietal Synchrony at Theta Band

Switching between tasks requires high-level cognitive control to instantiate task representation and prepare for motor response, which is fulfilled by broad communication in frontoparietal region (Figure 5A). Indeed, frontoparietal synchrony measured by PLV, was stronger in switch trials than repeat trials (F(1, 50) = 4.875, *p* = 0.032, η_p_^2^ = 0.089). There was no significant difference between groups (F(1, 50) = 2.498, *p* = 0.120, η_p_^2^ = 0.048) or interaction effect between condition and group (F(1, 50) = 0.269, *p* = 0.606, η_p_^2^ = 0.005) (Figure 5B). Moreover, patients with higher PRL showed decreased frontoparietal synchronisation difference between switch and repeat trials (Spearman’s Rho = −0.395, *p* = 0.046). This is further supported by dividing patients into subgroups based on PRL level. There was a significant interaction effect between subgroups and trial types (F(1, 24) = 4.421, *p* = 0.046, η_p_^2^ = 0.156) (Figure 5C). Whereas lower PRL groups showed stronger frontoparietal communication in switch trials than repeat trials (F(1, 24) = 8.20, *p* = 0.009), this pattern was absent among higher PRL groups (F(1, 50) = 0.01, *p* = 0.913) (Figure 5D). Hence, abnormally high PRL secretion disrupted frontoparietal communications involved in task switching.

## 4. Discussion

The primary goal of this study was to investigate the cognitive flexibility of patients with prolactinomas in task switching. Our results showed deteriorated cognitive flexibility in patients with modest larger switch costs than HCs. HCs exhibited strong frontal theta activity in switch trials and the theta power is associated with better switch performance. Nevertheless, patients showed generally weak frontal theta activity. We also found stronger frontoparietal synchronisation in switch trials than in repeat trials, reflecting the increased need of cognitive control manifested as the interregional communication. Importantly, there was a negative correlation between frontoparietal synchronisation and PRL concentration such that patients with higher PRL would show decreased frontoparietal synchronization. This finding was further supported by dividing patients into lower PRL groups and higher PRL groups. The lower PRL groups maintained stronger frontoparietal communication in switch trials, whereas the higher PRL groups did not show this pattern. Capitalized on multivariate decoding, we demonstrated that task-related selective brain patterns in switch and repeat trials were impaired in patients. This phenomenon resonates with the finding of frontal theta activity and together suggest patients’ inability to handle the context updating demand in switch trials. 

The behavioural results echoed previous findings that participants were slow in switch trials compared to the repeat trials [34,35]. Patients were also slower than HCs, additionally, patients demonstrated modest larger switch costs than HCs. The impaired cognitive flexibility of prolactinomas patients in our study is likely due to their deficits in differentiating the task type and task state. Previous studies showed that task switching activated the dorsolateral prefrontal cortex and anterior cingulate cortex [36,37] and switch trials evoke stronger theta activity over the frontal region than repeat trials during the cue interval period [38,39]. We observed stronger theta power in support of task switching among HCs. On the contrary, this phenomenon was absent in patients who showed generally weak theta activity. The negative correlation between theta activity and switch cost indicated that frontal theta activity is associated with better task switching performance as suggested in previous research [40,41]. Smaller switch costs are expected in participants with larger frontal theta power differences between switch and repeat trials. With the observation of reduced frontal gray matter volume in patients with prolactinomas [25], we speculate that the inability to exert the cognitive control undertaken by local frontal theta activity is a product of the anatomical changes. Consequently, patients showed weaker theta activity in switch trials and larger switch cost.

In terms of global network features, Lopez et al. reported stronger theta frontoparietal connectivity in switch trials than repeat trials during the cue interval [42]. We observed stronger frontoparietal synchronisation in switch trials than repeat trials, which validates previous research and supports the link between theta frontoparietal connectivity and proactive control [43,44]. Of note, patients with higher PRL manifested decreased frontoparietal synchronisation difference between switch and repeat trials. Whereas the lower PRL subgroups maintained the frontoparietal communication to cope with task switching, the higher PRL subgroups suffered from a destroyed frontoparietal synchrony. Therefore, abnormally high PRL secretion disrupted frontoparietal communications involved in task switching. 

Overproduction of PRL increases the number of cells secreting antibodies against myelin oligodendrocyte glycoprotein [45], a type of glial cell primarily provides support and insulation for axons in some vertebrates’ central nervous systems [46]. Abnormal PRL secretion could lead to a decrease in oligodendrocytes, ultimately impairing the protective and supportive functions of the cortical structure and the communication between different brain regions [47]. Moreover, PRL is expressed in distributed brain regions including the thalamus, cerebral cortex, hypothalamic, amygdala, etc. [48]. By virtue of the anti-correlation between dopamine secretion and prolactin production [49,50], PRL overproduction will disrupt the dopamine pathway and therefore impaired cognitive abilities [4,51]. These might account for why patients showed lower frontal theta power and frontoparietal connectivity. 

Machine-learning approaches enable the extraction of information from scalp EEG signals. A previous study utilizing a similar cued task-switching paradigm decoded task sets and lower-level stimulus characteristics [52]. Lower-level representations of the task cue and the task-relevant feature are prerequisites for task execution. Abstract task-set representations are also indispensable in configuring task states. Our results showed that the classification accuracy of task type in HCs was significantly higher than the chance level while the classification accuracy of patients was barely above chance. This suggests that the HCs manifested distinctive brain activation patterns in the colour task and the shape task. Analogously, the classification of switch trials over repeat trials was successful in HCs but not in patients. This implied that overproduction of PRL altered brain function and lead to generalized task coding under different conditions. Of note, multivariate decoding results also corroborated the finding that patients did not show a significant theta power increase in switch trials in the cue interval period. Together, these have indicated patients’ deficits in discriminating the task and state in a goal-directed manner.

There were several limitations in this research. Since we recorded the EEG at the scalp level, we cannot point out exactly which neuroanatomical regions are responsible for the task switching deficits caused by prolactinomas. Conjunct analyses combing various neuroimaging methods would contribute to high-resolution source reconstruction. Second, we excluded patients with big tumours to avoid their compression on surrounding tissues which would cause cognitive deficits. Future research should quantitively screen the tumour size using MRI and examine how tumour size would affect cognition on a fine scale. At present, there is still a lack of objective research evidence on the impact of treatment methods on the cognitive function of prolactinomas. It was found that surgery significantly improved the attention and memory of patients with pituitary adenoma (but still significantly different from normal controls), but which had no effect on impaired executive function. Therefore, future research can compare cognitive functions in patients after surgery and after drug treatment.

## 5. Conclusions

We investigated the neuropsychological mechanism of patients with prolactinomas in task switching. Prolactinomas patients experienced larger switch costs than HCs because of their deficits in differentiating the task type and task state as evidence by non-selective brain patterns throughout the colour and shape tasks as well as during the switch and repeat states. In addition, PRL overproduction alters frontoparietal synchronisation and aggravates switch costs. These findings proved the impaired executive functions in prolactinomas, particularly in the format of flexible task switching.

## Figures and Tables

**Figure 1 brainsci-13-00082-f001:**
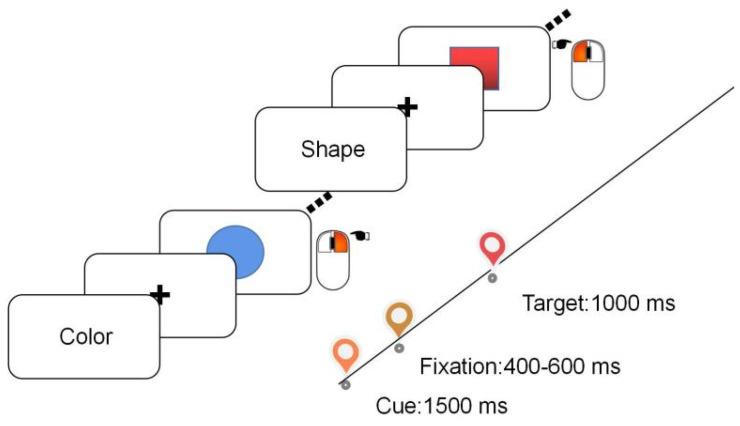
Schematic illustration of the colour-shape task-switching paradigm. Participants respond using a computer mouse.

**Figure 2 brainsci-13-00082-f002:**
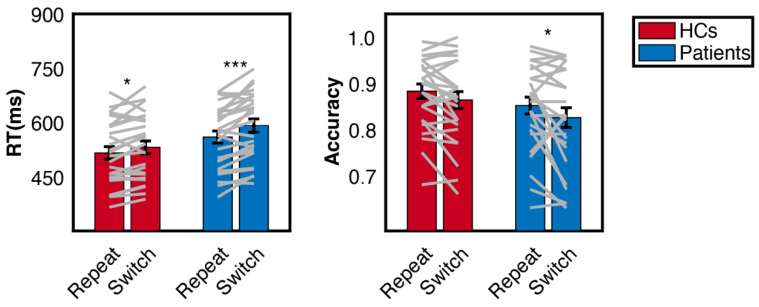
Reaction time and accuracy of patients and the HCs. Grey lines correspond to each participant. The errorbar represented the between-subject standard error. Asterisks indicate significance of simple effect analysis. * *p* < 0.05, *** *p* < 0.001.

**Figure 3 brainsci-13-00082-f003:**
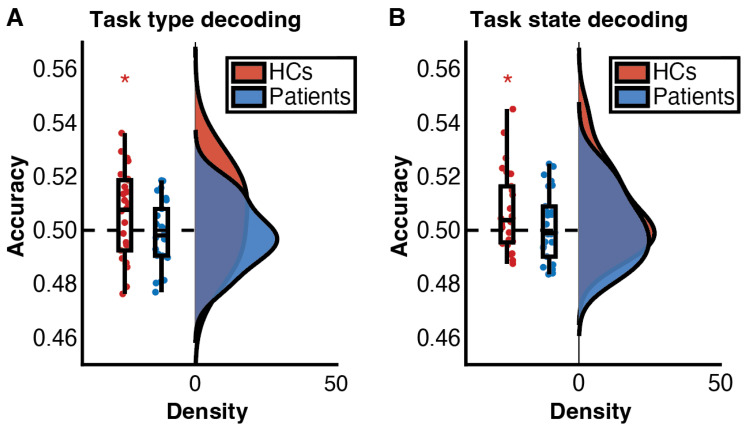
RainCloudPlot [33] of task type decoding (**A**) and task state decoding (**B**). The central line of the boxplot indicates the median of the data while the lower and upper boundary lines reflect the 25th and 75th percentile. Whiskers indicate the non-outlier range (1.5 × interquartile range). Coloured dots represent individuals’ decoding accuracies. The dashed horizontal line denotes the chance-level decoding accuracy. Violin plots show the probability distribution of the decoding accuracy. * *p* < 0.05.

**Figure 4 brainsci-13-00082-f004:**
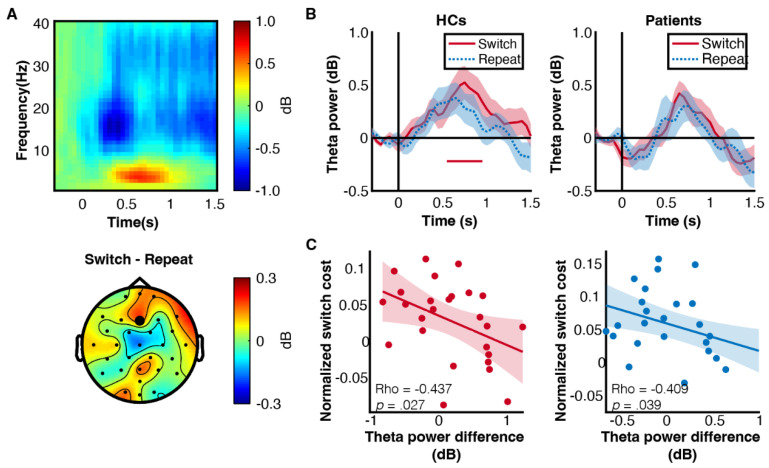
Frontal theta activity. (**A**) Time-frequency map of cue interval. Data was extracted from Fz and averaged across groups and conditions. The bottom panel is the topographical distribution of group averaged theta power difference between switch and repeat trials (2–7 Hz, 0.3–1 s). The black dot highlights the channel of interest. (**B**) Theta power curve of each group and each condition. The solid line below the x-axis is the significant time cluster. The shaded area is the between-subject standard error. (**C**) Spearman rank correlation between frontal theta activity and normalized switch cost. Theta activity is measured as the power difference between switch and repeat trials within the time window (0.3–1 s). Normalised switch cost is the RT difference between switch and repeat trials divided by their average. Normalisation can remove individual response speed difference. Coloured dots represent individuals’ data. The shaded area is the 95% confidence interval.

**Figure 5 brainsci-13-00082-f005:**
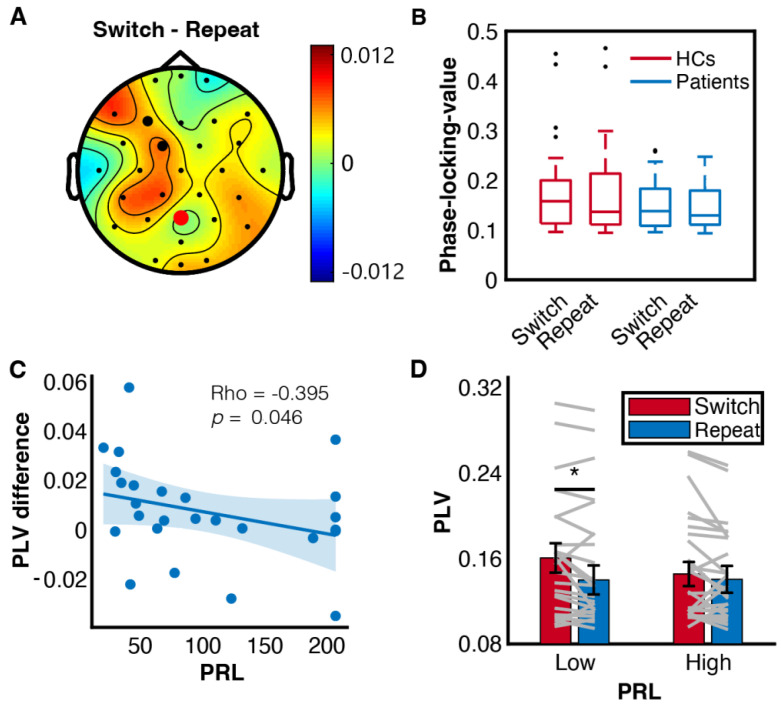
Frontoparietal connectivity at theta band. (**A**) topography of PLV referenced at Pz. FC1 and F3 were selected for frontoparietal phase synchrony analysis. (**B**) Boxplot of PLV in each condition. Central lines indicate the median, and the bottom and top edges of the box indicate the 25th and 75th percentiles. Black dots represent outliers. (**C**) Spearman rank correlation of PRL and PLV difference between switch and repeat trials. Coloured dots represent individuals. (**D**) Frontoparietal connectivity of patients subgrouped by PRL. * *p* < 0.05.

**Table 1 brainsci-13-00082-t001:** Demographic information of participants.

	Patients	HCs	Statistic
Gender	18 Females	13 Females	*χ*^2^ = 1.997, *p* = 0.158 ^a^
Age	34.3 (11.94, 18–58)	34.6 (10.52, 21–56)	*t*(50) = 0.086, *p* = 0.931 ^b^
Education	12.7 (2.28, 9–16)	14.0 (3.48, 6–20)	*t*(50) = 1.651, *p* = 0.105 ^b^

Standard deviation and range were provided in parentheses. ^a^ Chi-Square Tests, ^b^ two independent samples *t*-test.

## Data Availability

Data are available upon reasonable request.

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
