# Peer review of "Neuropsychological Alterations of Prolactinomas’ Cognitive Flexibility in Task Switching"

_brainsci, 2023, doi:10.3390/brainsci13010082_

Round 1
Reviewer 1 Report
The present study about impaired cognition related to prolactinomas is well written, structured, conducted and documented.
The authors shgowed that PRL high blood level alters frontoparietal synchronization resulting in impaired executive functions especially to regard flexible task switching.
In the present resarch study the authors investigated the cognitive flexibility and capability of patients with prolactinomas in task switching; the question is interesting although not new. This topic has already been explored but the authors by the present study try to link the functional relevance of frontal cortex impairment with cognitive deficits by exploring EEG theta activity. The methods and results sound well and the conclusion well support the results and initial hypothesis. On the other hand the authors do not analyze/discuss in detail if different patient profile could be found in relation to radiology/bioilogy/age/sex/EEG features in the contexte of prolactinoma (please add details, discuss); furthermore the authors do not discuss the implication of therapy on cognitive performance (please add details, discuss) and any possible future/different management strategies/perspectives.
Author Response
We are grateful for the suggestion. To be more clear and in accordance with the reviewer concerns, we have added a brief description as follows: At present, there is still a lack of objective research evidence on the impact of treatment methods on the cognitive function of prolactinomas. It was found that surgery significantly improved the attention and memory of patients with pituitary adenoma (but still significantly different from normal controls), but which had no effect on impaired executive function. Therefore, future research can compare cognitive functions in patients after surgery and after drug treatment.
Reviewer 2 Report
The authors in their article evaluated the neuropsychological mechanism of patients with prolactinomas in task switching. Pituitary adenomas are the second most common intracranial tumours and can generate seriously mental problems. Here scalp electroencephalography of participants was used during they were performing a colour-shape switching task. Results of in the study showed the impaired executive functions in patients with prolactinomas, particularly in the format of flexible task switching.
The manuscript is interesting because we are still lacking data concerning detailed disruptions in central nervous system caused by prolactinomas. The study should be continued with bigger patients cohort inclusion of MRI assessment of tumors, but this was also discussed by the authors.
Minor Issues:
1. Please check the length of abstract - should mi max 200 words.
2. Number of ethical confirmatyn made by the ethics committee should be given in line 112.
3. Figure 3 and 5 legends need to be feed with statistical descriptions - meaning of "*".
4. I encourage to prepare supplementary materials showing particular prolactin levels measured in patients included in the research.
Author Response
We are grateful for the suggestion. To be more clear and in accordance with the reviewer concerns, we have added a brief description as follows:
- Please check the length of abstract - should mi max 200 words.
- Nmeber of ethical confirmatyn made by the ethics committee should be given in line 112.
- Figure 3 and 5 legends need to be feed with statistical descriptions - meaning of "*".
- I encourage to prepare supplementary materials showing particular prolactin levels measured in patients included in the research.
Respond
1. We have shortened the abstract to less than 200 words.
2. We have added the number of ethical confirmation in line 113.
3. We have added the statistical descriptions in Figure 3 and Figure 5 legends.
4. The datasets for this manuscript are not publicly available at this moment because the data recording is ongoing for more subjects. Future requests to access the datasets should be directed to the corresponding author.
Reviewer 3 Report
This paper is well written and doesn't needs further revision.
Author Response
We appreciate the reviewer’s positive evaluation of our work.